# A Chitosan-Based Biomaterial Combined with Mesenchymal Stem Cell-Conditioned Medium for Wound Healing and Skin Regeneration

**DOI:** 10.3390/ijms242216080

**Published:** 2023-11-08

**Authors:** Filip Humenik, Ján Danko, Lenka Krešáková, Katarína Vdoviaková, Vladimír Vrabec, Emília Vasilová, Mária Giretová, Štefan Tóth, Zuzana Fagová, Ján Babík, Ľubomír Medvecký

**Affiliations:** 1Department of Morphological Sciences, University of Veterinary Medicine and Pharmacy in Košice, 041 81 Košice, Slovakia; 2Educational, Scientific and Research Institute AGEL, 811 06 Bratislava, Slovakia; 3Clinic of Birds, Exotic and Free Living Animals, University of Veterinary Medicine and Pharmacy in Košice, 041 81 Košice, Slovakia; 4Division of Functional and Hybrid Systems, Institute of Materials Research of SAS, 040 01 Košice, Slovakia; 5Department of Histology and Embryology, University of Pavol Jozef Šafárik, 041 80 Košice, Slovakia; 6Clinic of Burns and Reconstructive Medicine, AGEL Hospital, 040 15 Košice-Šaca, Slovakia

**Keywords:** biomaterial, wound healing, mesenchymal stem cell, conditioned medium

## Abstract

The aim of this study was to provide a beneficial treatment effect of novel chitosan bio-polymeric material enriched with mesenchymal stem cell products derived from the canine adipose tissue (AT-MSC) on the artificial skin defect in a rabbit model. For the objectivity of the regeneration evaluation, we used histological analysis and a scoring system created by us, taking into account all the attributes of regeneration, such as inflammatory reaction, necrosis, granulation, formation of individual skin layers and hair follicles. We observed an acceleration and improvement in the healing of an artificially created skin defect after eight and ten weeks in comparison with negative control (spontaneous healing without biomaterial). Moreover, we were able to described hair follicles and epidermis layer in histological skin samples treated with a chitosan-based biomaterial on the eighth week after grafting.

## 1. Introduction

Trauma, burns and chronic diseases can lead to the formation of refractory wounds, which in exceptional cases end fatally [1,2,3]. Therapy based on conventional procedures often fails. Procedures based on the use of tissue skin grafts or growth factors are gaining prominence. Among the latest procedures includes the use of mesenchymal stem cells (MSCs), their products being contained in a conditioned medium (CM) even in combination with biomaterial, which affects the whole spectrum of processes applied during the healing of chronic wounds—inflammatory reaction, proliferation and remodeling [4,5,6]. In recent years, attention has been directed to materials of natural origin with conductive ability or its high efficiency, processing flexibility and ease of manipulation and control. The abovementioned biomaterials are based on carbon nanomaterials, conductive polymers or metal-based biomaterials [7,8,9,10]. A separate area in wound healing and tissue regeneration is represented by chitosan-based biomaterials. Chitosan as a conductive natural compound exhibits antimicrobial activity, improves hemostatic effect, supports cell proliferation, and mediates complete wound regeneration and epithelial reconstruction [11,12,13]. MSCs play an important role in the wound healing process, especially in the inflammatory, proliferative and remodeling phases [14,15]. MSCs are characterized by their ability of immunomodulation, when, depending on the microenvironment in which they are located, they can reduce the proliferation of T-lymphocytes, the differentiation of B lymphocytes and affect the function of NK cells [16,17]. In the process of remodeling, MSCs release growth factors (VEGF, HGF, IGF, PDGF and TGF-ß), which contribute to the proliferation and migration of fibroblasts and keratinocytes, the growth and formation of new blood vessels and synthesis collagen and extracellular matrix proteins and at the same time prevent its degradation by inhibiting metalloproteinase [18,19]. The paracrine effect of MSC on keratinocytes remains unclear. To understand the paracrine MSC mechanism, it is necessary to map the transcriptome and proteome of the cells and their secreted molecules. With regard to the paracrine activity of MSC-affected wound healing, target cells at the wound site must express receptors that bind to secreted mediators. While the effects of VEGF on endothelial cells in angiogenesis were extensively studied, the role of VEGF receptors (VEGFR-1, VEGFR-2 and VEGFR-3) in keratinocytes is not so well defined [20,21,22]. In a previous study, we confirmed the angiogenic effect of bone marrow mesenchymal stem cell-conditioned medium (BMMSC-CM) on a chorioallantoic membrane (CAM) model, which was related to the secretion of VEGF and other pro-angiogenic factors [23]. VEGF is also thought to play a non-angiogenic role targeting keratinocyte migration and proliferation [24]. The use of CM compared to MSC has several indisputable advantages: minimization of unwanted recipient response, production in larger quantities, the possibility of influencing the quantitative and qualitative composition, and lower requirements for storage and transport [25].

The aim of this study was to create a chitosan-based biopolymeric scaffold and verify its effectiveness in combination with adipose tissue mesenchymal stem cell-conditioned medium (ATMSC-CM) on the wound healing process on an artificially created skin defect in a rabbit animal model.

## 2. Results

### 2.1. Characterization of MSC from Canine Adipose Tissue

MSC of isolated population showed spindle shape morphology and 120 × 15 µm. The yield of isolated cells from adipose tissue was 2.5 × 10^6^ cells/g. The results of flow cytometry show high positivity for CD29 (99.3 ± 0.7%), CD44 (99.1 ± 0.5%), CD90 (85.9 ± 0.6%) and CD105 (99.7 ± 1.2%) and negative for CD45 (1.0 ± 0.2%) (Figure 1). Using a commercial multilineage differentiation kit (StemPro Multilineage Differentiation Kit, Gibco, Paysley, Scotland), we were able to confirm high osteogenic and chondrogenic differentiation capability. However, the ability to differentiate for adipocytes was not confirmed (Figure 2).

### 2.2. Preparation and Characterization of Chitosan-Based Biopolymeric Scaffolds

The average molecular size (Mw) of biopolymers in scaffolds was 73 kD and 85 kDa for chitosan and polyhydroxybutyrate (PHB), respectively. The porosity of the scaffolds was around 90%. The fiber-like macroporous microstructure of the scaffold was found via SEM, where irregularly shaped large macropores up to 150 µm size as well as a high portion of micropores less than 20 µm size were visible. Moreover, long chitosan fibers with about 10 µm diameter and fine globular agglomerates of PHB can be identified in the microstructure (Figure 3B). Optical micrographs of lyophilized scaffold (Figure 3C) clearly demonstrate the presence of large macropores even ~200 μm size with a chitosan fiber crosslinked microstructure and globular agglomerates of PHB-precipitated biopolymer particles. The pore size distribution in substrates (Figure 4) determined using image analysis clearly verified a large fraction (about 70%) of smaller pores (<10 µm), and the portion of larger > 90 µm pores was approximately 10%. The mass of wet scaffold after water uptake achieved a 4.2 ± 0.3 time of origin of the dry mass of the samples, which clearly verified the strong hydrophilic character of blend and high porosity scaffolds.

### 2.3. Cell Seeding on the 3D Porous Polymeric Scaffolds and Cytototxicity Evaluation

Formazan production of fibroblasts in contact with the scaffolds for 1, 7 and 14 days of culture indicated an increase in the number of viable cells, which demonstrates that the scaffolds support cell growth (Figure 5A). Images from live/dead fluorescence staining (Figure 5B) documented that after 7 days of cultivation, there were almost no dead cells in the construct cross-sections (Figure 5B) and there was a dense population of live cells.

### 2.4. Animal Model

The macroscopic examination confirmed that artificial skin defect had healed well in all examined animals after the respective observation periods (8th, 10th and 12th week). It was observed that treatment with the CHIT/PHB biomaterial enriched with the conditioned medium caused faster wound closure than the control, similarly to the non-enriched biomaterial. Next, we observed that wounds treated with CHIT/PHB shows no hypertrophy of granulation tissue and formation of a scab, again as compared to the negative control (Figure 6(A1,A2,B1,B2)). However, after 12 weeks, all lesions were healed with newly formed fur, over the entire extent of the defect (Figure 6(A3,B3)).

### 2.5. Histological Study

The results of histological staining prove that after eight and ten weeks from the creation of the defect and the subsequent application of biomaterial and biomaterial enriched with the conditioned medium, the formation of granulation tissue and of hair follicles occurred (Figure 7(B1,B2,C1,C2), Appendix A). We observed the same result in histological samples examined after ten weeks, which in our histopathological score means 1/1- (Figure 7(B1,B2,C1,C2); Table 1). In the negative control samples from the same time period, we find necrotic lesions and absent epidermis, which we evaluate as 3 in our scoring system (Figure 5(A1,A2) and Appendix A; Table 1). However, after twelve weeks, we observe in the control and experimental group skin of a normal structure without the appearance of necrotic foci, with a formed epidermis and hair follicles. Such a finding corresponds to the value 0/1 in our scoring system (Figure 7(A3–C3); Table 1). The results of histological analysis and assessment of collected skin samples *n* = 2 from all group were the same or very similar.

## 3. Discussion

The main goal of the submitted study was to point out the possibility of using a biomaterial based on chitosan and next its combination with mesenchymal stem cell-conditioned medium to improve wound healing in vivo.

The adipose tissue MSC population used in this study was the same as the one we used previously, and the identical protocol for isolation of MSC-conditioned media was used [26,27]. The CD characterization of AT-MSC used in this study shows high positivity for CD29 (99.3 ± 0.7%), CD44 (99.1 ± 0.5%), CD90 (85.9 ± 0.6%) and CD105 (99.7 ± 1.2%), but negativity for CD45 (1.0 ± 0.2%). The expression of CD90 is important for the wound healing process as it promotes angiogenesis and accelerates wound closure, which is closely related to the perfusion of damaged tissue [28]. The equally high expression of CD105 promises a positive effect in the wound healing process, because its interaction with the receptor for pro-angiogenic TGF-β [29]. The expression of CD44 is connected with wound healing, too. As a previously published study described, the expression of CD44 mainly affects the inflammatory phase and fibrotic response of the wound healing process [30].

MSCs and their products can affect all phases of the wound healing process. In the inflammatory phase of the wound healing process, the MSCs exert immune-suppressive effects via the production of IL-4 and IL-10 and through TNF suppression, which is connected to the blocking of T-Cell proliferation [14,31,32,33,34,35]. Next, MSCs play a role in the proliferative phase via the secretion of VEGF, HGF, PDGF, TGF-β, KGF, CXCL12 and EGF, which can be isolated in conditioned medium. It can be connected to the recruitment of keratinocytes, fibroblasts, angiogenesis and neovascularization [36,37,38]. Finally, in the remodeling phase, MSCs attend via the production of numerous soluble factors and cytokines (HGF, PGE2, adrenomodulin, IL-10), which enhance the epithelial–mesenchymal transition and suppress myofibroblast differentiation [39,40]. The presence of MSCs in healthy skin and the previously described function of MSCs and their products in the wound healing process suggest that the application of exogenous MSCs is promising in the treatment of wounds and skin defects.

In this study, we experimentally used ATMSC-CM isolated from dog mesenchymal stem cells, which we applied to a rabbit animal model for wound healing. This is the first documented cross-species local application of CM under in vivo conditions in combination with a biomaterial for wound healing. The amount of protein (2.25 mg/mL) in the isolated conditioned medium was very similar to that described by other authors, even in CM isolated from a population of human MSCs [41,42]. Our study described efficacy of chitosan/PHB based biomaterial and chitosan/PHB biomaterial combined with canine ATMSC-CM on wound healing process in a rabbit animal model. Chitosan is widely used in wound healing trials, for its antibacterial, anti-inflammatory, low immunogenicity and biocompatibility properties. Similarly, chitosan shows the ability to enhance cell adhesion, migration and proliferation, and its high water absorption capacity, which maintains wound humidity and bacteriostatic properties. The combination of chitosan and PHB creates a material with improved mechanical strength and matrix porosity, which is important for nutrient diffusion and cell infiltration [43,44,45]. The results were confirmed via histological analysis and via the use of a histopathological scoring system that comprises attributes such as the dermis/epidermis architecture, organization of collagen fibers, presence of hemorrhage, inflammatory cells and hair follicles. We chose to create our own system, since an official system for histopathological evaluation of wound healing in rabbits is absent. However, the system takes into account the attributes of previous histopathological systems for wound healing that have been used in other animal models like mice and rats [46,47,48,49]. On the 8th and 10th week after implantation of biomaterial, was observed that treatment with the CHIT/PHB biomaterial enriched with the conditioned medium caused a faster wound closure than the control, which was the same as with the non-enriched biomaterial. Next, we observed that wounds treated with CHIT/PHB show no hypertrophy of granulation tissue and formation of a scab, again compared to the negative control. However, after 12 weeks, all lesions were healed with newly formed fur, over the entire extent of the defect. The results of histological staining correlate with clinical statue. It showed, that after eight and ten weeks from the creation of the defect and the subsequent application of chitosan-based biomaterial and the same biomaterial enriched with the conditioned medium, the formation of granulation tissue and of hair follicles occurs. We observed the same result in the histological samples examined after ten weeks, which in our histopathological score means 1/1-. Very similar results were declared by other authors who performed similar experiments in rat and mice models [50,51,52]. The results obtained using the enriched biomaterial seem to be better by half a point. It correlates to the results from our previously published study where MSC-CM was used [27,53,54].

## 4. Materials and Methods

First, we needed to obtain informed consent from the dogs’ owners, which is an essential criterion for approval of the study by the Ethical Committee of UVMP (EC UVMP) in Košice. The study was approved by EC UVMP on 2 September 2021 (EKVP/2021-01) and State Veterinary and Food Administration of the Slovak Republic (no. 1827/09-221/3).

### 4.1. Isolation and Characterization of Canine Amniotic MSC-Conditioned Medium Harvesting

The MSC population used in this experiment was the same as the one we used in our previous study [26].

#### 4.1.1. Isolation of MSC from Canine Adipose Tissue

Adipose tissue (AT) was harvested from purebred healthy dogs (*n* = 3): Donor 1—male, German Shepherd, 35 kg, 3 years old. Donor 2—female, Cane Corso, 70 kg, 4 years old. Donor 3—male, German Shorthaired Pointer, 32 kg. Before adipose tissue collection, all donors were examined (clinical examination, biochemical, and hematological parameters were evaluated). Samples of adipose tissue (5–7 g) were collected during surgical procedures under general anesthesia from the subcutaneous tissue in the scapular area. The tissue was then washed with phosphate-buffered saline (PBS; Biowest, Nuaillé, France) containing 2% ATB + ATM, then mechanically dissociated and enzymatically digested with 0.05% collagenase type I and IV (Sigma, Burlington, MA, USA) at 37 °C for 1 h. At the end of the incubation period, the digested tissue was filtered (through a 100 μm cell strainer) to remove tissue fragments, centrifuged at 400× *g* for 10 min, and the obtained stromal vascular fraction (SVF) pellet was resuspended in culture medium consisting of DMEM-F12 containing 10% FBS and 2% ATB + ATM and plated in a T25 tissue culture flask at a concentration of 106 cells/flask and incubated in cultivation medium (DMEM-F12 + 10% FBS + 2% ATB + ATM) at 37 °C and 5% CO_2_. After 48 h, non-adherent cells were removed, and subsequently, the medium was changed twice a week.

#### 4.1.2. Expression of Surface Markers

Samples were analyzed for positive mesenchymal stromal cell markers (CD29, CD44, CD90 and CD105) and for hematopoietic stem cells marker (CD45). Each sample was diluted to a final concentration of 2 × 10^5^ cells/mL and centrifuged at 400× *g*/5 min. Subsequently, the supernatant was removed and the cell pellet was resuspended in 100 μL of PBS containing 3–5 μL of CD90 ((YKIX337.217, allophycocyanin; APC), CD29 ((MEM-101, phycoerythrin; PE), CD44 (MEM-263, APC), CD105 ((MA1-19594, fluorescein isothiocyanate; FITC) and CD45 (YKIX716.13, PE)—all ThermoFisher, Waltham, MA, USA—and incubated for 60 min at 4 °C in the dark. At the end of the incubation period, the samples were centrifuged again at 400× *g*/5 min, the supernatant was removed and the sample was washed in 200–500 µL of washing solution (1% FBS in PBS + 0.1% Sodium Azide (SevernBiotech Ltd., Kidderminster, UK). Cytometric analysis was performed on a BD FACSCanto^®^ flow cytometer (Becton Dickinson Biosciences, San Jose, CA, USA) equipped with a blue (488 nm) and red (633 nm) laser and 6 fluorescence detectors. The percentage of cells expressing individual CD traits was determined via a histogram for the respective fluorescence. The data obtained via measurement were analyzed in BD FACS DivaTM analysis software v9.0. As a negative control, we used the same type of non-labelled MSCs for autofluorescence control.

#### 4.1.3. Preparation of AT-MSC-Conditioned Medium

The CM were prepared as described in our previous publication [27]. Shortly after, AT-MSC (P3) were cultured in MEM Alpha (Biowest) without FBS (Biowest). After 48 h incubation in a humidified atmosphere with 5% CO_2_ at 37 °C, collected media samples were filtered through a 0.2 µm sterile syringe filter (Millipore, Burlington, MA, USA). To ensure that equal concentrations (2.25 mg/mL) of CM were used for the subsequent experiments, the protein concentration of the CM was quantified via Bradford protein assay using standard Bradford reagent (Sigma). As a control (nonconditioned medium), MEM Alpha was regarded. Samples of ATMSC-CM were collected and stored at −80 °C until use.

### 4.2. Preparation and Characterization of Composite PHB/CHIT Scaffolds

The PHB/CHIT blend (the mass ratio of 1:1) was prepared via mixing of biopolymer solutions, where chitosan (Sigma–Aldrich, St. Louis, MO, USA) was dissolved in 1% (*w*/*v*) solution of acetic acid and PHB (GoodFellow, Cambridge, UK) in propylene carbonate. Solutions were mixed at a 1:1 ratio in a magnetic stirrer at 400 rpm. The biopolymer were co-precipitated by adding NH3aq (20% solution, Sigma–Aldrich, for High Performance Liquid Chromatography (HPLC)) followed filtration and washed with distilled water. The aqueous suspensions of precipitated biopolymers were molded into plastic form (3D printed, DaVinci, XYZ-printing, Lake Forest, CA, USA), polylactidacid polymer). The final substrates, after freezing at −20 °C, were lyophilized (Ilshin, Ede, The Netherlands) for 8 h.

Water uptake of scaffolds was measured in triplicate via immersion of lyophilized substrates (approximately 100 mg) in saline solution at 37 °C up to a constant mass, and swelling was calculated as the ratio of the wet-to-dry sample weights, and results were expressed as mean ± SD.

The microstructure of scaffolds was observed via scanning electron microscopy (FE 250 SEM JEOL7000) and by using optical microscope Nikon LV-DIA in transmission mode.

The true density of prepared blend after lyophilization was determined using a Helium Pycnometer (AccuPyc II, Micrometics) which was used for the calculation of the total porosity of scaffolds (from dimensions and mass). Pore size distribution was measured according to method described in our previous study [55].

The average molecular weights (Mw) of PHB and CHIT polymers were determined via gel permeation chromatography (GPC, Watrex, RI detector) on a PL gel mixed C 5 μm column (PHB) and PL gel mixed OH 8 μm (chitosan) with chloroform and 0.01 M 257 NaH_2_PO_4_ (pH = 5) solutions as the mobile phases for the characterization of PHB and chitosan, respectively, at a flow rate of 1 mL/min.

### 4.3. Cell Seeding on the 3D Porous Polymeric Scaffolds and Cytototxicity Evaluation

In vitro proliferation of PHB/CHIT blends was evaluated after seeding L929 mouse fibroblasts (ECACC, Salisbury, UK) according to EN ISO 10993-5:2009 [56]. The 19  ±  1 × 10^6^ cells/cm^3^ scaffolds were filtered over scaffolds via syringe. The number of untapped cells (counted in Neubauer hemocytometer) in culture media, which have penetrated over scaffold after cell suspension filtration via syringe, was around 4–5  ×  10^5^, from which it resulted that the seeding efficiency was about 90%. Scaffolds were cut with a scalpel into eight identical parts (volume of one construct was 25  ±  5 μL). The number of cells in one construct after cutting was 5  ±  0.2  ×  10^5^. Scaffolds were transferred separately into a well of 48 nonadherent culture plate (Greiner Bio-One, Kremsmünster, Austria) and incubated (37 °C, 5% CO_2_, 95% humidity) for 2 h. Following that, 0.5 mL of recommended cultivation medium was added (DMEM (Biowest) + 2 mM Glutamine (SAFC Biosciences, Hampshire, UK) + 10% FBS (Sigma-Aldrich, St. Louis, MO, USA) and ATB-ATM (penicillin, streptomycin, amphotericin) solution (Sigma-Aldrich, St. Louis, MO, USA)). The medium was changed three times a week.

The cell proliferation test was conducted via the commercially available Cell Titer Aqueus One Solution Cell Proliferation Assay (MTS, Promega) after 1, 7 and 14 days using a UV-VIS spectrophotometer (Shimadzu, Griesheim, Germany) at a wavelength of 490 nm.

The attachment and morphology of cells were visualized via live/dead fluorescence staining using fluorescein diacetate/propidium iodide solution (green, live cells and red, dead cells) after 7 days of cell cultivation using an inverted optical fluorescence microscope (Leica DM IL LED, blue filter).

### 4.4. In Vivo Model for Wound Healing and Surgical Technique

Twelve healthy adult rabbits of Hyla breed were used in this study. Ethical approval was obtained from the State Veterinary and Food Administration of the Slovak Republic no. 1827/09-221/3. The average weight of animals was 3300 g (range: 3000–3750 g), and at the time of surgery, the rabbits were from 12 to 16 months old (sex ratio = 1:1). General inhalation anesthesia was performed by using Isoflurane (Isofluran Piramal, Chiesi Pharmaceuticals GmbH, Vien, Austria) in concentration of 5% for the induction of anesthesia and 3.5% for surgical procedure. In all rabbits, the area around an impending operation on the back was shaved and prepared with a Betadine and alcohol solution using a sterile technique. We made a square artificial defect via the excision of skin and subcutaneous tissue. All animals were divided into 2 groups. Group 1 (*n* = 6): 2 skin lesions of 3 × 3 cm were created. The skin after excision enlarged its area. After that, the CHIT/PHB scaffold was implanted in the wound cranially and the CHIT/PHB was enriched with canine AT-MSC-conditioned medium in an amount of 2 mL/cm^2^ caudally. Group 2 (*n* = 6): skin lesions of 3 × 3 cm were created. After that, the CHIT/PHB scaffold was implanted in the wound cranially, and the caudal lesion was without scaffold, like the control (Figure 8A–C). Primary wound coverage was performed via sterile gauze pad fixed to intact skin using restorable suture material. The secondary layer of the outer covering was a waterproof and breathable film (Curapor, Lohman and Raucher int., Rengsdorf, Germany), and the tertiary layer and fixation of the bandage were ensured using a fixation elastic patch (Omnifix, Hartmann, Leer, Germany; Figure 8D–F). The wound dressings of 2nd and 3rd layer and control were made up every 72 h. Sampling for histological examination was carried out on 8th, 10th and 12th week after implantation. Surgical procedures for skin samples harvesting were performed in general anesthesia by using Isoflurane (Isofluran Piramal, Chiesi Pharmaceuticals GmbH, Vien, Austria) in a concentration of 5% for induction of anesthesia and 3.5% for surgical procedure. For histological study, we collected one sample (2 × 2 cm) from each animal (*n* = 2). After the samples were taken, the wound was closed and treated according to surgical standards. After recovery, the animals were used to teach clinical subjects.

### 4.5. Histological Processing

Skin biopsies were fixed in 4% paraformaldehyde solution and processed for routine histopathological examination. Paraffin sections (3–4 µm) were immersed in xylene and alcohol, stained via routine hematoxylin and eosin staining method, and rehydrated in alcohol and xylene. Slides were mounted in resin (Entellan; Merck, Germany). Two independent observers, blinded to the experimental groups, examined 6 sections for each animal and experimental group. The histopathological score was assessed separately by two of the investigators. An OLYMPUS BX50 light microscope (Olympus; Tokyo, Japan) with a CANON EOS 2000D digital camera (Canon; Tokyo, Japan) and QuickPHOTO Industrial 2.3 image analysis software (Promicra; Prague, Czech Republic) were used for semi-quantitative analysis. Each section was evaluated at 100× and 400× magnification.

### 4.6. Histopathological Score System for Wound Healing Evaluation

Standardized criteria for the quantitative assessment of skin wound healing have not yet been defined. Therefore, we used our own semi-quantitative scoring system (0–3) for the histopathological assessment of skin wound healing as follows: 0 = none; 1 = mild; 2 = moderate; 3 = severe. Each section was evaluated at 100× and 400× magnification.

Histopathological score for the evaluation of skin biopsies:

0 = none: Normal structure of skin with no histopathological lesions (well-defined architecture of epidermis and dermis, presence of hair follicles and other skin adnexa).

1 = mild: Structure resembles normal skin, small amount of immature tissue/granulation tissue in the dermis, collagen fibers organization is well defined, indistinct leukocyte infiltration and/or hemorrhage, hair follicles present, continuous epidermis without defects.

2 = moderate: More pronounced changes in skin structure, greater amount of immature/granulation tissue in the dermis, homogeneous eosinophilic staining of the extracellular matrix, more pronounced leukocyte infiltration and/or hemorrhage, absence of hair follicles, epidermis is present or/and altered (thickness and morphology) + discontinuous (focal defects are found).

3 = severe: Skin structure is unrecognizable, immature/granular tissue in the dermis, massive leukocyte infiltration and/or hemorrhage, necrotic lesions with calcification, absent skin adnexa, and absent or completely altered (discontinuous with numerous defects) epidermis.

## 5. Conclusions

The presented study describes the positive effect of a biomaterial based on chitosan in combination with canine adipose tissue stem cell products containing in conditioned medium on the wound healing process in a rabbit model. Mainly, the histological analysis of skin samples confirmed the positive influence of the enriched biomaterial as well as the biomaterial itself on the wound healing process. We observed almost complete healing already on the eighth and tenth week after implantation. Moreover, we partially confirmed the possibility of the cross-species local application of stem cell-conditioned medium, without observing any undesirable effects in the rabbit animal model. As mentioned above, we used the conditioned medium obtained from canine AT-MSC and applied it topically to an artificially created rabbit skin defect. The verification of this knowledge can help further progress and the use of stem cell products in the veterinary practice of small animals. On the other hand, it must be said that these results are only experimental and further research is needed. In the end, it is necessary to say that the therapeutic effect of the conditioned medium of stem cells depends on the content of soluble growing factors, EVs and free bioactive molecules, which are its essential parts.

## Figures and Tables

**Figure 1 ijms-24-16080-f001:**
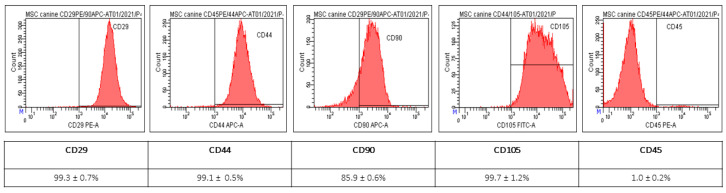
Results of CD analyses from AT-MSC from third passage (P3). The AT-MSC showed high positivity for CD29 (99.3 ± 0.7%), CD44 (99.1 ± 0.5%), CD90 (85.9 ± 0.6%) and CD105 (99.7 ± 1.2%) and low expression of CD45 (1.0 ± 0.2%).

**Figure 2 ijms-24-16080-f002:**
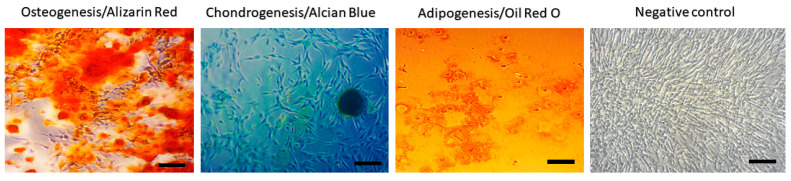
Multilineage potential of canine AT-MSC. Canine AT-MSC showed high osteogenic (presence of calcium deposits detected by Alizarin red) and chondrogenic potential (presence of glycoproteoglycanes detected by Alcian blue staining); however, cells showed low adipogenic potential (triglycerides detected by Oil Red O staining). Scale bars: 50 µm.

**Figure 3 ijms-24-16080-f003:**
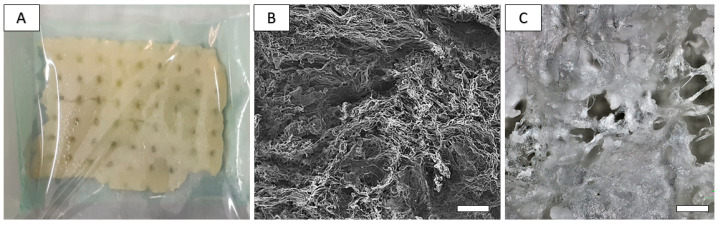
Appearance of composite PHB/CHIT scaffolds and microstructure. Macroscopically, the scaffold shows a sponge-like structure and is approximately 3–4 mm thick (**A**). The porosity of scaffolds was around 90%. The fiber-like macroporous microstructure of scaffold was found via SEM, where irregularly shaped large macropores up to 150 µm size ((**B**), scale bar: 100 µm) as well as high portion of micropores less than 20 µm size were visible ((**C**), scale bar: 20 µm).

**Figure 4 ijms-24-16080-f004:**
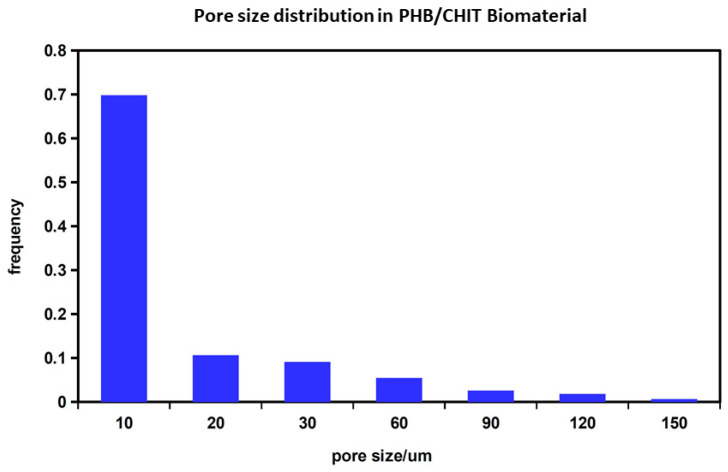
Pore size distribution determined via image analysis of optical micrographs. The pore size distribution in substrates determined using image analysis clearly verified a large fraction (about 70%) of smaller pores (<10 µm) and the portion of larger > 90 µm pores was approximately 10%.

**Figure 5 ijms-24-16080-f005:**
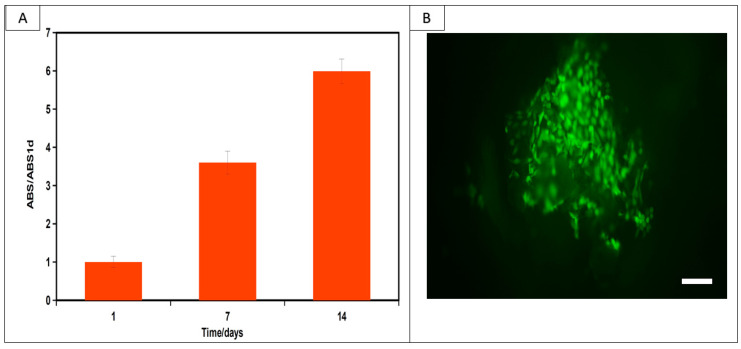
Viability of fibroblasts cultured on PHB/CHIT substrates determined via MTS test (**A**) and live/dead staining of cells after 7 days of cultivation (**B**). (**A**) demonstrates the continuous rise in proliferation of fibroblasts (cultured in MEM at 37 °C, 5% CO_2_ and 96% humidity) with culture time and live/dead cell staining; (**B**) indicates only the presence of spindle-shaped live cells on the surface of the substrate without the visible presence of dead cells that could identify some cytotoxicity of the samples. Scale bar: 75 µm.

**Figure 6 ijms-24-16080-f006:**
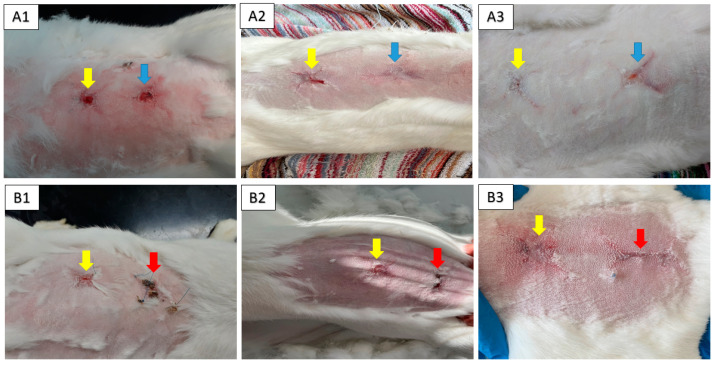
Wound healing progress during the experimental study. Photo documentation of healing process after biomaterial (yellow arrow), biomaterial enriched with conditioned medium (blue arrow) and negative control (red arrow) and wound healing progress at 8th week (**A1**,**B1**), 10th week (**A2**,**B2**) and 12th week (**A3**,**B3**).

**Figure 7 ijms-24-16080-f007:**
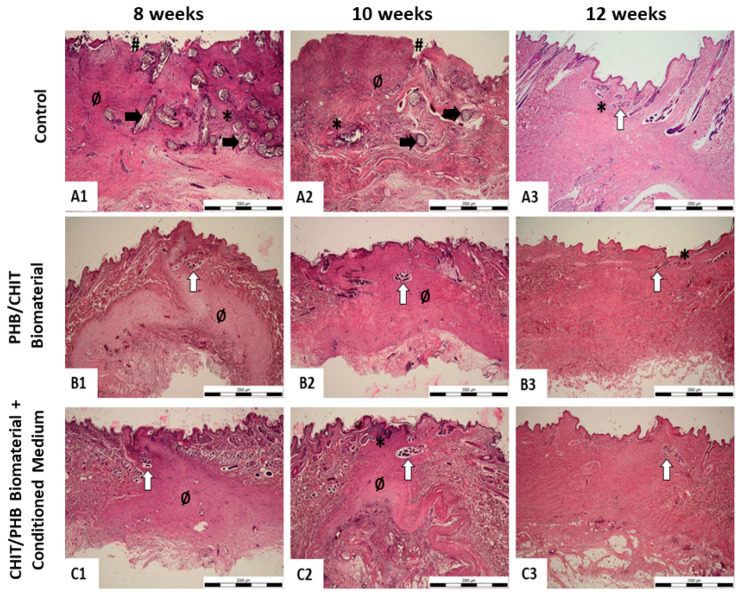
Representative microphotographs of histopathological analysis for each experimental group and healing period. (**A1**–**A3**) control group; (**B1**–**B3**) implant group; (**C1**–**C3**) implant+medium group; 1–8 weeks, 2–10 weeks, 3–12 weeks; (**A1**,**A2**): severe injury with necrotic lesions and absent epidermis (histopathological score 3), (**A3**): normal structure of skin with indistinct infiltration (histopathological score 0-); (**B1**,**B2**): small amount of immature/granulation tissue with hair follicle ingrowth (histopathological score 1-); (**B3**): normal structure of skin with indistinct infiltration (histopathological score 0-); (**C1**): small amount of immature/granulation tissue with hair follicle ingrowth (histopathological score 1); (**C2**): small amount of immature/granulation tissue with hair follicle ingrowth and indistinct infiltration (histopathological score 1-); (**C3**): normal structure of skin (histopathological score 0); black arrow: necrotic lesions; #: absent epidermis; *: leukocyte infiltration; white arrow: hair follicle ingrowth; Ø: immature/granulation tissue. Scale bar = 2000 µm.

**Figure 8 ijms-24-16080-f008:**
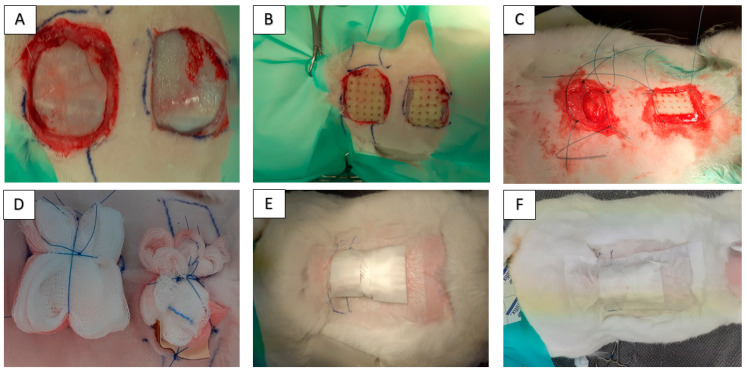
Photo documentation of the operative procedure. Creating an artificial skin defect across all layers of the skin (**A**). Implantation of CHIT/PHB biomaterial and CHIT/PHB enriched with ATMSC-CM on lesions in group 1 (**B**) and CHIT/PHB biomaterial and control (without implanation) on lesions in group 2 (**C**). Primary wound coverage performed using sterile gauze fixed by restorable suture material (**D**). The secondary layer of the outer covering (**E**) was waterproof and breathable film (Curapor, Lohman and Raucher int., Germany), and the tertiary layer and fixation of the bandage (**F**) were ensured using fixation elastic patch (Omnifix, Hartmann, Leer, Germany).

**Table 1 ijms-24-16080-t001:** Results of histopathological assessment of skin wound healing. Wounds treated with CHIT/PHB biomaterial (implant) and CHIT/PHB biomaterial enriched with canine ATMSC-CM (implant+medium) reached score range 1/1- at the 8th and 10th week. On the other hand, non-treated wounds reached score 3 at the same time. After 12 weeks, all wounds reached score 0/0- (control and implant + medium) or 0-/1 (implant). The number of samples taken in each group *n* = 2.

Experimental Group	Healing Period		
	8 Weeks	10 Weeks	12 Weeks
Control	3	3	0/0-
Implant	1-	1-	0-/1
Implant + medium	1/1-	1-	0/0-

## Data Availability

The data presented in this study are available upon request from the corresponding author.

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
