# Peer review of "A Chitosan-Based Biomaterial Combined with Mesenchymal Stem Cell-Conditioned Medium for Wound Healing and Skin Regeneration"

_ijms, 2023, doi:10.3390/ijms242216080_

Round 1

Reviewer 1 Report

Comments and Suggestions for Authors

In this manuscript, the authors developed a chitosan-based biomaterial and tested its efficacy for wound healing in a rabbit skin defect model. The biomaterial was tested both with and without conditioned media prepared from canine adipose tissue derived mesenchymal stem cells. An untreated control was also included. The results indicate that the biomaterial (both with and without the conditioned medium) helped accelerate the wound healing process.

I have the following questions and comments:

(1.) In line 59 on page 2, the acronym “CAM” has been used, but this has not been spelled out. Please spell out the full form of an acronym the first time you use it.

(2.) Subsection 2.1 of the Results section (lines 70 to 81, page 2) discusses flow cytometry results. However, the method has not been described in the Materials and Methods section. Please include a detailed description of the flow cytometry method in the Materials and Methods section.

(3.) In line 90 on page 3, the acronym “PHB” is introduced for the first time, but it is not spelled out. Please spell it out.

(4.) In lines 97 to 99, the authors state that “The mass of wet scaffold after swelling achieved approximately fourfold of origin dry mass of samples which clearly verified strong hydrophilic character of blend and high porosity scaffolds.”. Since the measurement was done in triplicate (lines 247 to 249, page 7), please report the exact increase for each of the three replicates in the form of a table.

(5.) Figures 4 and 5 (page 4) show representative images for the macroscopic wound healing and the histological results. Please show the images for all the animals included in this study in a Supplementary Information document.

(6.) Table 1 (page 5) shows the histological scores, but only a single number is reported for each group. Is this an average for all the animals included in each group? If yes, please clarify whether all the animals in each group had the same score? If they did not have the same score, please include the range of scores that was observed. Furthermore, please specify the number of animals included in each group in the caption for the table.

(7.) In the Discussion section (lines 169 to 180, page 5), the authors mention the various factors (IL4, IL10, VEGF etc.) that could be released by MSCs which could help with the wound healing process. Were any of these measured in the conditioned medium (say by using a multiplex ELISA)? If yes, please include the results. If not, please could you elaborate on why the CM composition was not characterized?

(8.) Please provide a brief explanation for why a canine model was used for harvesting MSCs, and why a rabbit model was used for studying the wound healing effects (either the Introduction section or the Materials and Methods section could be a good place to describe this).

(9.) In subsection 4.1 (lines 222 to 225, page 6), the authors only provide a citation for the MSC isolation and characterization process, as well as the CM characterization. Even if this has been described in a previous publication, please include a brief description here as well.

(10.) In lines 281-282 on page 8, the authors mention that “Sampling for histological examination was carried out on 8-th, 10-th and 12-th week after implantation.”. How many samples were taken at each time point? How were the samples harvested? Were the animals euthanized or not? If the animals were euthanized, what was the procedure for this? Please include all these details in this section.

(11.) This comment is a follow up to comment 7 above. In the Conclusions section (lines 330 to 333, page 9), the authors mention “In the end, it is necessary to say that the therapeutic effect of the conditioned medium of stem cells depends on the content of soluble growing factors, EVs and free bioactive molecules, which are its essential part.”. However, the content of the soluble factors/EVs/bioactive molecules have not been reported for the conditioned media. Were these quantified? If yes, please report the method details and the results. If not, please provide an explanation in the manuscript for why these were not measured.

Comments on the Quality of English Language

There are some errors in the sentence structures, but overall, the manuscript was understandable.

Author Response

Revision 1

Reviewer 1

Dear Reviewer,

Please find enclosed a revised manuscript with point-by-point responses to your comments.
We are thankful for your valuable and constructive comments and suggestions. We appreciate the time and effort that you dedicated to providing valuable feedback on our manuscript. We have
incorporated the changes to reflect most of your suggestions. Modifications made in the revised manuscript are marked up using the “Track Changes” on.

In this manuscript, the authors developed a chitosan-based biomaterial and tested its efficacy for wound healing in a rabbit skin defect model. The biomaterial was tested both with and without conditioned media prepared from canine adipose tissue derived mesenchymal stem cells. An untreated control was also included. The results indicate that the biomaterial (both with and without the conditioned medium) helped accelerate the wound healing process.

I have the following questions and comments:

  • In line 59 on page 2, the acronym “CAM” has been used, but this has not been spelled out. Please spell out the full form of an acronym the first time you use it.

Acronym „CAM“ was spelled according  to requirement. Please see line 59.

  • Subsection 2.1 of the Results section (lines 70 to 81, page 2) discusses flow cytometry results. However, the method has not been described in the Materials and Methods section. Please include a detailed description of the flow cytometry method in the Materials and Methods section.

Methodical procedure of flow cytometry according to requirements. Please see the lines 268-284.

  • In line 90 on page 3, the acronym “PHB” is introduced for the first time, but it is not spelled out. Please spell it out.

Acroynm „PHB“ was speeled according to requirement. Please the line 91.

  • In lines 97 to 99, the authors state that “The mass of wet scaffold after swelling achieved approximately fourfold of origin dry mass of samples which clearly verified strong hydrophilic character of blend and high porosity scaffolds.”. Since the measurement was done in triplicate (lines 247 to 249, page 7), please report the exact increase for each of the three replicates in the form of a table.

We are very sorry. This part of the study was carried out for us by an external workplace of the Slovak Academy of Sciences. After submitting your justified comment, however, they only sent us a modification of the sentences to the form: "The mass of wet scaffold after water uptake achieved 4.2 ± 0.3 time of origin dry mass of samples which clearly verified strong hydrophilic character of blend and high porosity scaffolds." Please see the lines 101-103. and for methodology: „Water uptake of scaffolds was measured triplicate by immersion of lyophilized substrates (approximately 100 mg) in saline solution at 37 °C up to a constant mass and swelling was calculated as the ratio of the wet to dry sample weights and results were expressed as mean±SD. See the lines 308-311. I hope that despite the incomplete completion of your request, you will be lenient.

  • Figures 4 and 5 (page 4) show representative images for the macroscopic wound healing and the histological results. Please show the images for all the animals included in this study in a Supplementary Information document.

As in the previous case, we cannot completely fulfill your comments. Since we did not foresee the need for additional photos of animals, we created only those that are mentioned in the manuscript. From the histological analyses, we attach extended panels for each group of animals. Please see suplementary materials : Figure S1-S3.

  • Table 1 (page 5) shows the histological scores, but only a single number is reported for each group. Is this an average for all the animals included in each group? If yes, please clarify whether all the animals in each group had the same score? If they did not have the same score, please include the range of scores that was observed. Furthermore, please specify the number of animals included in each group in the caption for the table.

The results of the histological analysis and evaluation were the same or very similar for the selected samples from the same group. (wording supplemented in the text, please see the lines 156-157). Likewise, the numbers of examined samples are added in the description of the table. Please see the line 176.

  • In the Discussion section (lines 169 to 180, page 5), the authors mention the various factors (IL4, IL10, VEGF etc.) that could be released by MSCs which could help with the wound healing process. Were any of these measured in the conditioned medium (say by using a multiplex ELISA)? If yes, please include the results. If not, please could you elaborate on why the CM composition was not characterized?

The conditioned medium used in this study was evaluated only quantitatively, not qualitatively. In our previous study, we used a detailed method (LC-MS/MS) of proteomic analysis  of the conditioned medium isolated from bone marrow stem cells for qualitative evaluation. However, this method is expensive, time-consuming (we waited almost a year for the results) and we performed it in cooperation with foreign partners. We do not know the qualitative structure of the given conditioned media, but we assume a similarity with the already analyzed conditioned media.

  • Please provide a brief explanation for why a canine model was used for harvesting MSCs, and why a rabbit model was used for studying the wound healing effects (either the Introduction section or the Materials and Methods section could be a good place to describe this).

We decided for dogs as donors of adipose tissue for further isolation mainly due to the fact that dogs are the most frequent patients in small animal veterinary practice and therefore can be used as a tissue donor under certain circumstances. On the other hand, we would not allow ourselves to experimentally application of the MSC-CM differently than in an approved experiment (in our case on rabbits). This explanation in modified version was added to Conclusion. Please, see the lines 427-432.

  • In subsection 4.1 (lines 222 to 225, page 6), the authors only provide a citation for the MSC isolation and characterization process, as well as the CM characterization. Even if this has been described in a previous publication, please include a brief description here as well.

Methodology of MSC isolation and Characterization was added. Please set he lines 251-266.

  • In lines 281-282 on page 8, the authors mention that “Sampling for histological examination was carried out on 8-th, 10-th and 12-th week after implantation.”. How many samples were taken at each time point? How were the samples harvested? Were the animals euthanized or not? If the animals were euthanized, what was the procedure for this? Please include all these details in this section.

The details of sample collection and next procedures were added. Please, see the lines 370-376.

  • This comment is a follow up to comment 7 above. In the Conclusions section (lines 330 to 333, page 9), the authors mention “In the end, it is necessary to say that the therapeutic effect of the conditioned medium of stem cells depends on the content of soluble growing factors, EVs and free bioactive molecules, which are its essential part.”. However, the content of the soluble factors/EVs/bioactive molecules have not been reported for the conditioned media. Were these quantified? If yes, please report the method details and the results. If not, please provide an explanation in the manuscript for why these were not measured.

Unfortunately, as we mentioned above, we did not perform a qualitative analysis of the given conditioned media. This statement is there because the reviewers insisted on it in the previous publication.

Reviewer 2 Report

Comments and Suggestions for Authors

1. In figure 1, English letters and numbers need to be unified font.

2. The author mentioned that the mass of wet scaffold after swelling achieved approximately fourfold of origin dry mass of samples, whether there is to water retention experiments, and the specific results of the experiment are best to show.

3. Figure 5 should be handled in a way that the angle at which the pictures were taken is consistent. The scale or uniform reference should be indicated to better compare the wound healing situation. Background for wound healing can be strengthened by citing 10.1002/adma.202306632; 10.1016/j.ccr.2023.215426

4. For figure 6, it would be better to indicate the date and group directly above or to the left of the figure. It would be better not to label A1, A2 next to the picture and then explain it in the figure notes.

5. In the animal experiment, based on the pictures shown by the authors, it is not obvious that the experimental group healed better than the control group. After 12 weeks, all the lesions healed and new fur was formed, why the designed material was used? What are the advantages of the designed material?

6. The article lacks data on in vitro aspects. The designed material is used for biological purposes and its biocompatibility should be determined first.

7. The author mentions that bone marrow mesenchymal stem cells can influence all stages of the wound healing process, and the exact mechanism should be experimentally verified and demonstrated.

Author Response

Revision 1

Reviewer 2

Dear Reviewer,

Please find enclosed a revised manuscript with point-by-point responses to your comments.
We are thankful for your valuable and constructive comments and suggestions. We appreciate the time and effort that you dedicated to providing valuable feedback on our manuscript. We have
incorporated the changes to reflect most of your suggestions. Modifications made in the revised manuscript are marked up using the “Track Changes” on.

  1. In figure 1, English letters and numbers need to be unified font.

English letters and numbers  were unified according requirement. Please see the Fig.1.

  1. The author mentioned that the mass of wet scaffold after swelling achieved approximately fourfold of origin dry mass of samples, whether there is to water retention experiments, and the specific results of the experiment are best to show.

We are very sorry. This part of the study was carried out for us by an external workplace of the Slovak Academy of Sciences. After submitting your justified comment, however, they only sent us a modification of the sentences to the form: "The mass of wet scaffold after water uptake achieved 4.2 ± 0.3 time of origin dry mass of samples which clearly verified strong hydrophilic character of blend and high porosity scaffolds." Please see the lines 101-103. and for methodology: „Water uptake of scaffolds was measured triplicate by immersion of lyophilized substrates (approximately 100 mg) in saline solution at 37 °C up to a constant mass and swelling was calculated as the ratio of the wet to dry sample weights and results were expressed as mean±SD. See the lines 308-311. I hope that despite the incomplete completion of your request, you will be lenient.

  1. Figure 5 should be handled in a way that the angle at which the pictures were taken is consistent. The scale or uniform reference should be indicated to better compare the wound healing situation. Background for wound healing can be strengthened by citing 10.1002/adma.202306632; 10.1016/j.ccr.2023.215426

Yes, we completely agree with this statement. However, the photo documentation was carried out by the auxiliary staff of the clinic. We cannot supply you with other photos. In the same way, the addition of the scale would probably disturb the already not quite satisfactory quality. As in the previous case, we cannot completely fulfill your comments. From the histological analyses, we attach extended panels for each group of animals. Please see suplementary materials : Figure S1-S3.

Both citation were added. Please see the lines 475-480.

  1. For figure 6, it would be better to indicate the date and group directly above or to the left of the figure. It would be better not to label A1, A2 next to the picture and then explain it in the figure notes.

Figure 6 was modified according the requirement. Please see Fig. 7.

  1. In the animal experiment, based on the pictures shown by the authors, it is not obvious that the experimental group healed better than the control group. After 12 weeks, all the lesions healed and new fur was formed, why the designed material was used? What are the advantages of the designed material?

Likewise, we agree with your statement. That is why we performed a histological analysis and evaluation of the samples taken, which showed us that wounds treated with a combination of biomaterial and CM or biomaterial alone heal faster (8-10 weeks = score 1/1- compared to negative control samples at the same time, almost 3)

  1. The article lacks data on in vitro The designed material is used for biological purposes and its biocompatibility should be determined first.

Data from in vitro study was added according your requirement. Please see the lines 117-128 and 325-346.

  1. The author mentions that bone marrow mesenchymal stem cells can influence all stages of the wound healing process, and the exact mechanism should be experimentally verified and demonstrated.

Again, I have to agree, but this statement is supported by the works of other authors who are cited in the given work.

Reviewer 3 Report

Comments and Suggestions for Authors

Overall the introduction endorse a very brief of state of art rather then investigation in depth of literature. This is a very vast studied field therefore it is not possible to cover in details this field by just 13 references.

The clarity of figures should be improved, especially the font sizes used for the images is poor now. For example the images from figure 1make almost impossible the reading of what is plotted for X and Y axis

“Using commercial...” please provide details which kit did you have used

“The porosity of scaffolds..” how you have measured the porosity ? pleas provide in details specification for each method used in this research

Please describe in details of methods used for the investigation used as not clear how you put or implemented the skins in what you claim in Figure 4

“The results of histological staining prove that after eight and ten weeks..” some citation are required

Table 1 some citations are required

The conclusion are very brief and do not cover and express actually the quantitative results gathered in this study

Comments on the Quality of English Language

 some imrpovement is required

Author Response

Revision 1

Reviewer 3

Dear Reviewer,

Please find enclosed a revised manuscript with point-by-point responses to your comments.
We are thankful for your valuable and constructive comments and suggestions. We appreciate the time and effort that you dedicated to providing valuable feedback on our manuscript. We have
incorporated the changes to reflect most of your suggestions. Modifications made in the revised manuscript are marked up using the “Track Changes” on.

Overall the introduction endorse a very brief of state of art rather then investigation in depth of literature. This is a very vast studied field therefore it is not possible to cover in details this field by just 13 references.

We fully agree and that's why we added another 12 references. Please see the lines 29-65.

The clarity of figures should be improved, especially the font sizes used for the images is poor now. For example the images from figure 1make almost impossible the reading of what is plotted for X and Y axis

Same, we have to fully agree, but it is the problem of flow cytometer software that we have been dealing with for a long time, but so far unsuccessfully.

“Using commercial...” please provide details which kit did you have used

Specification was added according requirement. Please see the line 76.

“The porosity of scaffolds..” how you have measured the porosity ? pleas provide in details specification for each method used in this research

We added a description of almost all the methods used in the given study, eventually we added a citation to our previous work in which the given methodology was used. Please see the "materials and methods" section

Please describe in details of methods used for the investigation used as not clear how you put or implemented the skins in what you claim in Figure 4

Since macroscopically we did not observe significant differences in the healing defects, for better orientation and confirmation of the hypothesis, we we performed a histological analysis and evaluation of the samples taken, which showed us that wounds treated with a combination of biomaterial and CM or biomaterial alone heal faster (8-10 weeks = score 1/1- compared to negative control samples at the same time, almost 3)

“The results of histological staining prove that after eight and ten weeks..” some citation are required

Ciation was added according to requirement. Please see the line 149.

Table 1 some citations are required

Table one is cited in study. Please see the line 151, 153, 156.

The conclusion are very brief and do not cover and express actually the quantitative results gathered in this study

Conclusion was modified, Please see the line 420-434.

Round 2

Reviewer 3 Report

Comments and Suggestions for Authors

.

Comments on the Quality of English Language

.